# MoGenTS: Motion Generation based on Spatial-Temporal Joint Modeling

**Weihao Yuan**[1][*]  **Yisheng He**[1][*]  **Weichao Shen**[1]  **Yuan Dong**[1]  **Xiaodong Gu**[1]
**Zilong Dong**[1][†]  **Liefeng Bo**[1]  **Qixing Huang**[2]
[1] Alibaba Group    [2] The University of Texas at Austin

## Abstract

Motion generation from discrete quantization offers many advantages over continuous regression, but at the cost of inevitable approximation errors. Previous methods usually quantize the entire body pose into one code, which not only faces the difficulty in encoding all joints within one vector but also loses the spatial relationship between different joints. Differently, in this work we quantize each individual joint into one vector, which i) simplifies the quantization process as the complexity associated with a single joint is markedly lower than that of the entire pose; ii) maintains a spatial-temporal structure that preserves both the spatial relationships among joints and the temporal movement patterns; iii) yields a 2D token map, which enables the application of various 2D operations widely used in 2D images. Grounded in the 2D motion quantization, we build a spatial-temporal modeling framework, where 2D joint VQVAE, temporal-spatial 2D masking technique, and spatial-temporal 2D attention are proposed to take advantage of spatial-temporal signals among the 2D tokens. Extensive experiments demonstrate that our method significantly outperforms previous methods across different datasets, with a 26.6% decrease of FID on HumanML3D and a 29.9% decrease on KIT-ML.

## 1   Introduction

Human motion generation from textual prompts is a fast-growing field in computer vision and is valuable for numerous applications like film making, the gaming industry, virtual reality, and robotics. Given a text prompt describing the motion of a person, the goal is to generate a sequence containing the positions of all joints in the human body at each moment corresponding to the text prompt.

Previous attempts to tackle this challenge generally proceeded in two directions. The first directly regresses the continuous human motions from the text inputs using methods such as generative adversarial networks (GANs) [1], variational autoencoders (VAEs) [2, 3, 4, 5], or recent diffusion models [6, 7, 8, 9, 10, 11]. Although continuous regression has the advantage of directly optimizing towards ground-truth data and does not lose the numerical precision, it struggles with the challenge of regressing continuous motion that encompasses intricate skeletal joint information and is limited by the quality and scale of current text-to-motion datasets. The second leverages the vector quantization (VQ) technique to convert continuous motion to discrete tokens, which transform the regression problem into a classification problem [12, 13, 14, 15, 16, 17]. In this way, the difficulty of motion generation could be greatly reduced. In a broader picture, approaching continuous regression problems using quantization techniques is becoming increasingly popular [18, 19, 20]. In this paper, we seek to push the limit of the second path by designing a novel representation and learning paradigm.

---

[*]Equal Contribution    † Corresponding Author
Project Page: `https://aigc3d.github.io/mogents`

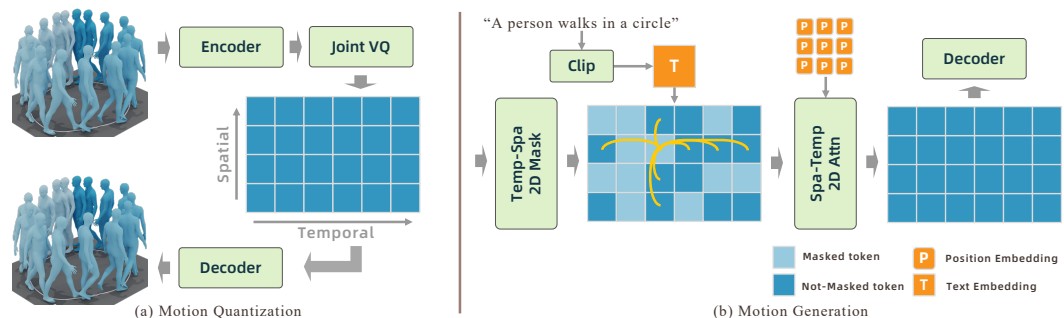

Figure 1: Framework overview. (a) In motion quantization, human motion is quantized into a spatial-temporal 2D token map by a joint VQ-VAE. (b) In motion generation, a temporal-spatial 2D masking is performed to obtain a masked map, and then a spatial-temporal 2D transformer is designed to infer the masked tokens.

Although recent methods have achieved impressive results by taking advantage of motion quantization [13, 14, 17, 21], they share an inherent drawback. The VQ process inevitably introduces approximation errors, which impose undesirable limits on the quality of the generated motions. Therefore, improving the accuracy of the VQ approximation is a key point in these methods. To this end, many techniques are proposed, such as residual VQ-VAE [17] and hierarchical VQ-VAE [22], to improve quantization precision and have obtained commendable results.

However, almost all previous methods quantize all joints of one frame into one vector and approximate this vector with one code from the codebook. This is not optimal since the whole-body pose contains much spatial information, i.e. the positions of all joints, such that quantizing the entire pose into one vector possesses two drawbacks. The first one is that this makes the encoding process difficult, as each code within the codebook is tasked with encapsulating the comprehensive information of all joints, making the quantization fundamentally more complex. The second one lies in the loss of spatial relationships between the individual joints, hence the subsequent network could not capture and aggregate the spatial information.

To address these problems, we propose to quantize each joint rather than the whole-body pose into one vector. This brings three benefits. First, encoding at the joint level significantly simplifies the quantization process, as the complexity associated with representing the information of a single joint is markedly lower than that of the entire pose. Second, with each joint encoded separately, the resulting tokens maintain a spatial-temporal distribution that preserves both the spatial relationships among joints and the temporal dynamics of their movements. Third, the spatial-temporal distribution of these tokens naturally organizes into a 2D structure, akin to that of 2D images. This similarity enables the application of various 2D operations, such as 2D convolution, 2D positional encoding, and 2D attention mechanisms, further enhancing the model's ability to interpret and generate human motion effectively.

In this paper, starting from the 2D motion quantization, we propose a spatial-temporal modeling framework for human motion generation. We employ a spatial-temporal 2D joint VQVAE [23] to encode each joint across all frames into discrete codes drawn from a codebook, which results in 2D tokens representing a motion sequence, as shown in Figure 1. Taking advantage of the 2D structure, both the encoder and decoder are equipped with 2D convolutional networks for efficient feature extraction, similar to 2D images. Then we perform the masked modeling technique following language tasks [24] and some prior motion generation works [17, 21]. However, unlike previous methods, we propose a temporal-spatial 2D masking strategy tailored for handling the 2D tokens. Then the randomly masked tokens are predicted by a spatial-temporal 2D transformer conditioned on the text input. The spatial and temporal locations of different tokens are first encoded by a 2D position encoding, after which the 2D tokens are processed by both the spatial and temporal attention mechanisms. The spatial-temporal attention considers not only whether the generated motion conforms to the input text in terms of temporal sequence, but also whether the generated joints are sound in terms of spatial structure. Extensive experiments across different datasets demonstrate the efficacy of our method in both motion quantization and motion generation. Compared to the previous SOTA method, the FID is decreased by 26.6% on HumanML3D and 29.9% on KIT-ML.

The main contributions of this work are summarized as follows:

- We novelly quantize the human motion to spatial-temporal 2D tokens, where each joint is quantized to an individual code of a VQ book. This not only makes the quantization task more tractable and reduces approximation errors, but also preserves crucial spatial information between individual joints.

- The 2D motion quantization enables the deployment of 2D operations analogous to 2D images, such that we introduce 2D convolution, 2D position encoding, and 2D attention to enhance the motion auto-encoding and generation.

- We propose a temporal-spatial 2D masking strategy and perform attention in both the temporal and spatial dimensions, which ensures the quality of the motion in both the temporal movement and the spatial structure.

- We outperform previous methods in both motion quantization and motion generation.

## 2 Related Work

### 2.1 Motion Generation from Continuous Regression

The initial methods devised for human motion generation were predominantly focused on directly regressing continuous motion values, a rather straightforward approach without intermediary quantization [2, 3, 25, 26, 27, 28]. Numerous strategies have utilized the variational autoencoder (VAE) framework to merge the latent embeddings of encoded text with those of encoded poses, subsequently decoding this combined representation into motion predictions [2, 3, 26, 4, 5]. Others have explored the potential of the recurrent network [25, 29, 30], generative adversarial network [28, 31, 1], or transformer network [32, 33, 34, 35] to improve the quality of motion regression. Drawing on the success of diffusion models, recent methods have begun to introduce the diffusion process into motion diffusion [36, 6, 7, 8, 9, 10, 11, 37, 38, 39, 40], yielding commendable results. To improve the diffusion model in human motion generation, various mechanisms are proposed to be integrated into the denoising process. A hybrid retrieval mechanism considering both semantic and kinematic similarities is integrated to refine the denoising process in [10]. Physical constraints [37] and guidance constraints [41] are also incorporated into the diffusion process for more reasonable motions. In addition, a module of construction supported by linguistics structure that constructs accurate and complete language characteristics and a module of progressive reasoning aware of the context are designed in [38].

While these methods hold the advantage of directly optimizing towards the ground truth with no information loss, they struggle with the challenge of regressing continuous motion that encompasses intricate skeletal joint information. Unlike the image tasks, there is no large-scale dataset for motion joints pre-training, therefore learning the continuous regression from scratch is not easy.

### 2.2 Motion Generation from Quantized Classification

To alleviate the difficulty of human motion generation, some methods quantize the continuous motion to discrete tokens and let networks predict the indices of the tokens, in which the original regression problem is converted to a classification problem [12, 13, 14, 15, 16, 17, 42, 22]. Typically, the input motion undergoes initial encoding via a VQ-VAE [23], which generates motion tokens for subsequent prediction. The prediction of the token indices can be approached in many ways. Drawing inspiration from advances in natural language processing, certain techniques adopt auto-regressive frameworks that predict tokens sequentially [13, 14, 42]. Others implement generative masked modeling strategies inspired by BERT [24], where tokens are randomly masked during training for the model to predict [17, 21]. The discrete diffusion is also introduced to denoise discrete motion tokens [15, 43]. More recently, large language models (LLMs) have been used to take over the prediction process [12, 44]. Multimodal control signals are first quantized into discrete codes and then formulated in a unified prompt instruction to ask the LLMs to generate the motion response [12].

Although all of these approaches reap the rewards of converting the task to a quantized classification paradigm, they also grapple with the inherent approximation error from quantization. Many techniques like EMA [14], code reset [14], and residual structure [45, 17] are introduced to improve quantization accuracy. However, the common practice of representing an entire human pose with a single token is fraught with difficulty and fails to preserve the spatial relationships across joints. In contrast, our approach takes a more granular path by quantizing each joint within the human skeleton

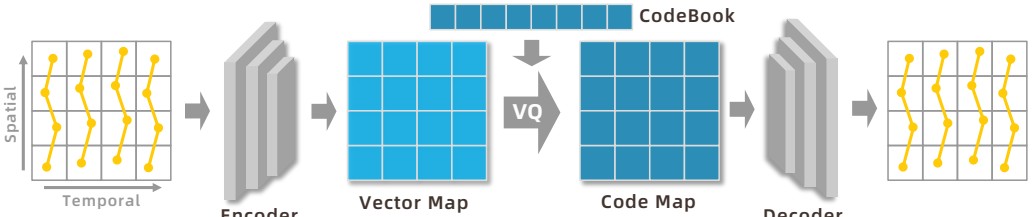

Figure 2: The structure of our spatial-temporal 2D Joint VQ-VAE for motion quantization.

independently. This not only makes the quantization task more tractable and reduces approximation errors, but also preserves crucial spatial information between the joints.

## 3 Method

### 3.1 Problem Statement

**Motion Quantization.** Given a motion sequence $\{\mathbf{M}_t\}_{t=1}^{T}$, the objective is to construct an optimal codebook $\mathbf{C} = \{\mathbf{c}_i\}_{i=1}^{C}$ capable of accurately representing this sequence, so that the encoded motion features $\mathbf{V}$ could be substituted by the corresponding codes $c_i$ from the codebook with the minimum loss, resulting in a discrete token sequence $\{\mathbb{O}\}$, representing the input motion.

**Motion Generation.** Given the textual prompt $\mathbf{T}$, the task of our framework is to generate the indices of the tokens $\{\mathbb{O}\}$, so that the vectors corresponding to these tokens could be decoded into a motion sequence aligned with the input text.

### 3.2 Overview

The overview of our method is illustrated in Figure 1. Our framework could be divided into two parts, motion quantization and motion generation. In the quantization phase, we first represent the input motion sequence in the joint-time structure, and encode it into a 2D vector map. Subsequently, the vectors are quantized into codes of a codebook, represented by the indices of the selected codebook entries, i.e., the joint tokens. Therefore, after the quantization, the input motion sequence is converted to tokens arranged in a 2D structure, where one dimension is spatial while the other one is temporal. This results in a 2D motion map, which is similar to a 2D image. In the generation phase, we mask the 2D token map with a temporal-spatial 2D masking strategy, and then use a 2D transformer to predict the masked tokens, conditioned on the CLIP [46] embedding $\mathbf{T}$ of the given text prompt. The 2D motion transformer considers both spatial attention and temporal attention between different 2D tokens. The 2D position embedding $\mathbf{P}$ is also used to convey the spatial and temporal locations of each token. Finally, the generated tokens are decoded back into a motion sequence that aligns with the given textual prompt, thus completing the process of text-driven motion generation.

### 3.3 Spatial-temporal 2D Joint Quantization of Motion

Directly regressing the continuous motion sequence is not easy. Thus, we perform motion quantization following previous methods on the whole pose [5, 17] or body part [42, 47, 48] to convert the regression problem to a classification problem, but differently, we quantize each joint rather than the whole-body pose. Therefore, we organize the motion sequence in a 2D structure and process it as a 2D map, as illustrated in Figure 2. For a motion with sequence length $T$ and joint number $J$, we employ a 2D convolutional network to encode the motion joints $\mathbf{J} = \{\mathbf{j}_t^j\}_{t=1,\ldots,T}^{j=1,\ldots,J}$ into 2D vectors $\mathbf{V} = \{\mathbf{v}_t^j\}_{t=1,\ldots,T}^{j=1,\ldots,J}$, where each vector $\mathbf{v}_t^j$ corresponds to time $t$ and joint $j$, as

$$\{\mathbf{v}_t^j\} = \mathcal{E}\left(\{\mathbf{j}_t^j\}\right), \tag{1}$$

where $\mathcal{E}$ denotes the encoder network composed of two convolutional residual blocks, and $\mathbf{j}_t^j$ denotes a 3D rotation angle of joint $j$ at time $t$. Then we quantize the vector with a codebook, i.e. we replace

the vector $\mathbf{v}_t^j$ with its nearest code entry $\tilde{\mathbf{v}}_t^j$ in a preset codebook $\mathbf{C} = \{\mathbf{c}_i\}_{i=1}^C$, as

$$\{\tilde{\mathbf{v}}_t^j\} = \mathcal{Q}(\{\mathbf{v}_t^j\}), \quad \mathcal{Q} : \tilde{\mathbf{v}}_t^j = \mathbf{c}_i \text{ where } i = \arg\min_i ||\mathbf{c}_i - \mathbf{v}_t^j||_2. \tag{2}$$

Here $\mathcal{Q}$ denotes the quantization process. After the quantization, the decoder $\mathcal{D}$ decodes the approximate vectors $\tilde{\mathbf{v}}_t^j$ to get the original joint information, as

$$\{\tilde{\mathbf{j}}_t^j\} = \mathcal{D}(\{\tilde{\mathbf{v}}_t^j\}) = \mathcal{D}(\mathcal{Q}(\mathcal{E}(\{\mathbf{j}_t^j\}))). \tag{3}$$

Finally, the auto-encoding network is optimized by a loss considering both the vector approximation and the decoded joints, as

$$\mathcal{L}_{vq} = \sum_{t,j} ||\tilde{\mathbf{j}}_t^j - \mathbf{j}_t^j||_1 + \alpha||\tilde{\mathbf{v}}_t^j - \mathbf{v}_t^j||_2, \tag{4}$$

where $\alpha$ denotes a weighting factor. We also employ the residual VQ-VAE structure, exponential moving average, and codebook reset following previous methods [14, 17], but for simplicity we only describe our method in single-level quantization without incorporating a residual structure in this section.

## 3.4   Temporal-spatial 2D Token Masking

After quantization of the motion joints, we obtain a 2D token map $\{\mathbb{O}_t^j\}$ as shown in Figure 3. We follow language models [24, 49] to perform random masking, but differently, here our tokens are in a 2D structure and there are totally $T \times J$ tokens. In this case, performing a 1D masking strategy ignoring the spatial-temporal relationship is not optimal, which applies the same masking strategy to all tokens. For example, it rarely appears that all joints in one frame are masked in the meantime. Networks trained in this way cannot work well in generating new motions, since the generation requires starting from an all-masked motion.

To solve this problem, we propose a 2D temporal-spatial masking strategy. We first perform the masking in only the temporal dimension and randomly mask the frames in the sequence. Once one frame is masked, all joints in this frame are masked, as illustrated in Figure 3 (a). After the temporal masking, we perform spatial masking on the remaining unmasked frames. Specifically, we randomly do the second masking in the spatial dimension for all joints of an unmasked frame. All masked tokens are replaced by the $\mathbb{O}_{\text{mask}}$ token, which denotes that this location is masked and predicted. Note that there is an underlying graph structure among the joints and that we have tried graph-based masking strategies. However, we find that the simple random masking strategy works the best. One interpretation is that it helps encode long-range correlations of joints in human motions, c.f. [50].

The temporal and spatial masking strategies adopt the same mask ratio schedule following [51, 52]. This ratio is computed as

$$\gamma(\tau) = \cos(\frac{\pi\tau}{2}), \tag{5}$$

where $\tau \in [0, 1]$. In training, $\tau \sim \mathbf{U}(0, 1)$ is uniformly sampled, leading to a mask ratio $\gamma(\tau)$. Then according to this ratio, $\gamma(\tau) \times T \times J$ tokens are randomly selected to be masked in temporal masking, and $\gamma(\tau) \times J$ tokens are randomly selected to be masked for one frame in the spatial masking. To add more disturbance to the masked prediction, the remasking mechanism in BERT [24] is also employed: If a token $\mathbb{O}_t^j$ is selected to be masked, it would be replaced with $\mathbb{O}_{\text{mask}}$ with an 80% probability, would be replaced with a random token $\mathbb{O}_i$ with a 10% probability, and remains unchanged with a 10% probability. After the temporal-spatial masking, we obtain a masked 2D token map for subsequent networks to predict.

## 3.5   Spatial-temporal 2D Motion Generation

### 3.5.1   Spatial-Temporal Motion Transformer

Given a text input and a masked token map, we use a transformer network to predict the tokens at the masked locations. There are three attentions performed in this transformer: spatial-temporal 2D attention, spatial attention, and temporal attention, as illustrated in Figure 3(b).

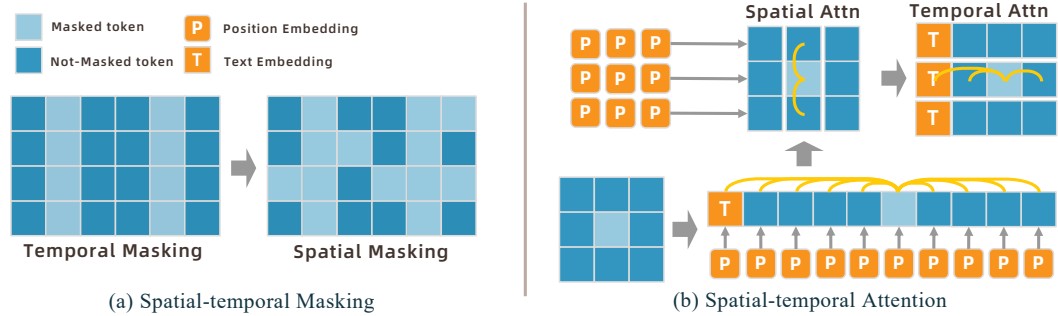

Figure 3: The temporal-spatial masking strategy (a) and the spatial-temporal attention (b) for motion generation.

**Spatial-Temporal 2D Attention.** For the spatial-temporal attention, we first extract the text feature embedding using CLIP [46], resulting in a text token $\mathbb{O}_{\text{text}}$. Then we add a 2D position encoding [53] to the 2D token map. The position encoding $\mathbf{P}_t^j$ is computed by the sinusoidal function in the spatial dimension $j$ and in the temporal dimension $t$, respectively, so it provides both the spatial position and the temporal position information for the subsequent attention network. After the 2D position embedding, we flatten the 2D token map to the 1D structure. Then we perform the 1D attention in the flattened spatial-temporal dimension, where the spatial and temporal relationship is reserved by the 2D position encoding, as

$$\mathcal{A}_{s-t} = \text{SoftMax}(Q \cdot K/\sqrt{d} + \mathbf{P})V, \tag{6}$$

where $Q, K, V \in \mathbb{R}^{(JT+1)\times d}$ are the query, key, and value matrices, d is the feature dimension, and $JT + 1$ is the number of tokens. This bidimensional attention is performed in a long sequence of $JT + 1$ tokens. Its strength is that it is able to learn patterns in the full spatial-temporal space. The limitation is that the mix of spatial and temporal information is too complex to learn. The resulting patterns are inaccurate.

**Joint Spatial Attention.** Joint spatial attention is the one-dimensional attention in only the spatial dimension. As shown in Figure 3(b), we ignore the temporal dimension and rearrange the 2D tokens. The tokens of different time frames are regarded as different batches, and then the attention is only calculated between different joints on each time batch, as

$$\mathcal{A}_s = \text{SoftMax}(Q_s \cdot K_s/\sqrt{d} + \mathbf{P})V_s, \tag{7}$$

where $Q_s, K_s, V_s \in \mathbb{R}^{J\times d}$. This spatial attention equally works on the joints of any time frame, so it could capture the inner relationship between different joints without the influence of temporal information. The advantage of this attention module is that it is easier to learn. However, it only guarantees the rationality of the different joints in one pose, regardless of whether the input text is or what the target motion is.

**Joint Temporal Attention.** Similar to spatial attention, we also perform one-dimensional attention in only the temporal dimension. As shown in Figure 3(b), we ignore the spatial dimension and rearrange the 2D tokens. The tokens of different joints are regarded as different batches, and then the attention is only calculated between different times on each joint batch, as

$$\mathcal{A}_t = \text{SoftMax}(Q_t \cdot K_t/\sqrt{d} + \mathbf{P})V_t, \tag{8}$$

where $Q_s, K_s, V_s \in \mathbb{R}^{T\times d}$. This temporal attention works equally on the time sequence of any joint. Similar to joint spatial attention, this attention module is easier to learn. However, it mostly captures the change in motion of the joint and guarantees the rationality of the movement of one joint, such as motion smoothness.

**Integration of attention modules.** Our approach applies spatial-temporal 2D attention, joint spatial attention, and joint temporal attention in a sequential manner. Here the spatial-temporal 2D attention provides the initialization, and joint spatial attention and joint-temporal attention offer regualarization.

### 3.5.2 Training and Inference.

**Training.** In the training, we use the spatial-temporal motion transformer to predict the probability logits of all $C$ tokens $\hat{y}_i$ in the masked positions. Then a cross-entropy loss is computed between the prediction $\hat{y}$ and the target $y$, as

$$\mathcal{L}_{\text{mask}} = -\sum_{\text{mask}} \sum_{i=1}^{C} y_i \log(\hat{y}_i) + (1 - y_i) \log(1 - \hat{y}_i). \tag{9}$$

We also employ the residual structure following previous methods [17], and use a residual spatial-temporal motion transformer to improve motion accuracy. The structure of the residual transformer is the same as that of the mask transformer, including spatial-temporal attention, spatial attention, and temporal attention. In training, we randomly select one layer $l$ from the $L$ residual layers to learn. All tokens in the preceding layers $[0 : l]$ are summed as the input of the residual transformer. The prediction of the $l$-th layer tokens is also optimized by the cross-entropy loss.

**Inference.** In the inference, the generation starts from an empty token map, i.e., all tokens are masked. The prediction of the masked tokens is repeated by $N$ iterations. In each iteration, we build the probability distribution of $C$ tokens from the SoftMax of the predicted logits, and sample the output from the distribution. In the next iteration, according to the schedule shown in Section 3.4 with $\tau = n/N$, low-score tokens are selected to be masked and predicted again, until $n$ reaches $N$. Once the iterative masked generation is finished, the residual transformer progressively predicts the residual tokens of the residual VQ layers. Afterward, the base tokens plus the residual tokens are decoded by the VQVAE decoder to get the final generated motion.

**Classifier Free Guidance.** The classifier free guidance (CFG) technique [54] is utilized to incorporate the text embeddings into the transformer architecture. During the training phase, the transformer undergoes unconditional training with a probability of $10\%$. In the inference, CFG is applied at the final linear projection layer just before the softmax operation. Here, the final logits, denoted as logits, are derived by moving the conditional logits $\text{logits}_{\text{con}}$ away from the unconditional logits $\text{logits}_{\text{un}}$ using a guidance scale $s$, as

$$\text{logits} = (1 + s) \cdot \text{logits}_{\text{con}} - s \cdot \text{logits}_{\text{un}}, \tag{10}$$

where $s$ is set to 4.

## 4 Experiments

### 4.1 Datasets

We evaluate our text-to-motion model on HumanML3D [5] and KIT-ML [55] datasets. HumanML3D consists of 14616 motion sequences and 44970 text descriptions, and KIT-ML consists of 3911 motions and 6278 texts. Following previous methods [5], 23384/1460/4383 samples are used for train/validation/test in HumanML3D, and 4888/300/830 are used for train/validation/test in KIT-ML. The motion pose is extracted into the motion feature with dimensions of 263 and 251 for HumanML3D and KIT-ML respectively. The motion feature contains global information including the root velocity, root height, and foot contact, and local information including local joint position, velocity, and rotations in root space. The local joint information corresponds to 22 and 21 joints of SMPL [56] for HumanML3D and KIT-ML respectively.

### 4.2 Implementation details

Our framework is trained on two NVIDIA A100 GPUs with PyTorch. The batch size is set to 256 and the learning rate is set to 2e-4. To quantize the motion data into our 2D structure, we restructure the pose in the datasets to a joint-based format, with the size of $12 \times J$. The data is then represented by the joint VQ codebook comprised of 256 codes, each with a dimension of 1024. Due to the global information included in the HumanML3D data representation, we also incorporate an additional global VQ codebook of the same size to encode the global information. For the residual structure, the number of residual layers is set to 5 following previous methods [17]. Both the encoder and decoder

| Methods | FID ↓ | Top1 ↑ | Top2 ↑ | Top3 ↑ | MM-Dist ↓ | Diversity → |
|---|---|---|---|---|---|---|
| Ground Truth | $0.002^{\pm.000}$ | $0.511^{\pm.003}$ | $0.703^{\pm.003}$ | $0.797^{\pm.002}$ | $2.974^{\pm.008}$ | $9.503^{\pm.065}$ |
| TEMOS [4] | $3.734^{\pm.028}$ | $0.424^{\pm.002}$ | $0.612^{\pm.002}$ | $0.722^{\pm.002}$ | $3.703^{\pm.008}$ | $8.973^{\pm.071}$ |
| TM2T [13] | $1.501^{\pm.017}$ | $0.424^{\pm.003}$ | $0.618^{\pm.003}$ | $0.729^{\pm.002}$ | $3.467^{\pm.011}$ | $8.589^{\pm.076}$ |
| T2M [5] | $1.087^{\pm.021}$ | $0.455^{\pm.003}$ | $0.636^{\pm.003}$ | $0.736^{\pm.002}$ | $3.347^{\pm.008}$ | $9.175^{\pm.083}$ |
| MDM [7] | $0.544^{\pm.044}$ | - | - | $0.611^{\pm.007}$ | $5.566^{\pm.027}$ | $9.559^{\pm.086}$ |
| MLD [6] | $0.473^{\pm.013}$ | $0.481^{\pm.003}$ | $0.673^{\pm.003}$ | $0.772^{\pm.002}$ | $3.196^{\pm.010}$ | $9.724^{\pm.082}$ |
| MotionDiffuse [8] | $0.630^{\pm.001}$ | $0.491^{\pm.001}$ | $0.681^{\pm.001}$ | $0.782^{\pm.001}$ | $3.113^{\pm.001}$ | $9.410^{\pm.049}$ |
| PhysDiff [37] | 0.433 | - | - | 0.631 | - | - |
| MotionGPT [12] | 0.567 | - | - | - | 3.775 | 9.006 |
| T2M-GPT [14] | $0.141^{\pm.005}$ | $0.492^{\pm.003}$ | $0.679^{\pm.002}$ | $0.775^{\pm.002}$ | $3.121^{\pm.009}$ | $9.761^{\pm.081}$ |
| M2DM [15] | $0.352^{\pm.005}$ | $0.497^{\pm.003}$ | $0.682^{\pm.002}$ | $0.763^{\pm.003}$ | $3.134^{\pm.010}$ | $9.926^{\pm.073}$ |
| Fg-T2M [38] | $0.243^{\pm.019}$ | $0.492^{\pm.002}$ | $0.683^{\pm.003}$ | $0.783^{\pm.002}$ | $3.109^{\pm.007}$ | $9.278^{\pm.072}$ |
| AttT2M [16] | $0.112^{\pm.006}$ | $0.499^{\pm.003}$ | $0.690^{\pm.002}$ | $0.786^{\pm.002}$ | $3.038^{\pm.007}$ | $9.700^{\pm.090}$ |
| DiverseMotion [43] | $0.072^{\pm.004}$ | $0.515^{\pm.003}$ | $0.706^{\pm.002}$ | $0.802^{\pm.002}$ | $2.941^{\pm.007}$ | $9.683^{\pm.102}$ |
| ParCo [42] | $0.109^{\pm.005}$ | $0.515^{\pm.003}$ | $0.706^{\pm.003}$ | $0.801^{\pm.002}$ | $2.927^{\pm.008}$ | $9.576^{\pm.088}$ |
| MMM [21] | $0.080^{\pm.003}$ | $0.504^{\pm.003}$ | $0.696^{\pm.003}$ | $0.794^{\pm.002}$ | $2.998^{\pm.007}$ | $9.411^{\pm.058}$ |
| MoMask [17] | $0.045^{\pm.002}$ | $0.521^{\pm.002}$ | $0.713^{\pm.002}$ | $0.807^{\pm.002}$ | $2.958^{\pm.008}$ | - |
| Ours | $\mathbf{0.033}^{\pm.001}$ | $\mathbf{0.529}^{\pm.003}$ | $\mathbf{0.719}^{\pm.002}$ | $\mathbf{0.812}^{\pm.002}$ | $\mathbf{2.867}^{\pm.006}$ | $9.570^{\pm.077}$ |
| Ground Truth | $0.031^{\pm.004}$ | $0.424^{\pm.005}$ | $0.649^{\pm.006}$ | $0.779^{\pm.006}$ | $2.788^{\pm.012}$ | $11.080^{\pm.097}$ |
| TEMOS [4] | $3.717^{\pm.028}$ | $0.353^{\pm.002}$ | $0.561^{\pm.002}$ | $0.687^{\pm.002}$ | $3.417^{\pm.008}$ | $10.84^{\pm.100}$ |
| TM2T [13] | $3.599^{\pm.153}$ | $0.280^{\pm.005}$ | $0.463^{\pm.006}$ | $0.587^{\pm.005}$ | $4.591^{\pm.026}$ | $9.473^{\pm.117}$ |
| T2M [5] | $3.022^{\pm.107}$ | $0.361^{\pm.005}$ | $0.559^{\pm.007}$ | $0.681^{\pm.007}$ | $3.488^{\pm.028}$ | $10.72^{\pm.145}$ |
| MDM [7] | $0.497^{\pm.021}$ | - | - | $0.396^{\pm.004}$ | $9.191^{\pm.022}$ | $10.85^{\pm.109}$ |
| MLD [6] | $0.404^{\pm.027}$ | $0.390^{\pm.008}$ | $0.609^{\pm.008}$ | $0.734^{\pm.007}$ | $3.204^{\pm.027}$ | $10.80^{\pm.117}$ |
| MotionDiffuse [8] | $1.954^{\pm.062}$ | $0.417^{\pm.004}$ | $0.621^{\pm.004}$ | $0.739^{\pm.004}$ | $2.958^{\pm.005}$ | $11.10^{\pm.143}$ |
| MotionGPT [12] | 0.597 | - | - | - | 3.394 | 10.54 |
| T2M-GPT [14] | $0.514^{\pm.029}$ | $0.416^{\pm.006}$ | $0.627^{\pm.006}$ | $0.745^{\pm.006}$ | $3.007^{\pm.023}$ | $10.86^{\pm.094}$ |
| M2DM [15] | $0.515^{\pm.029}$ | $0.416^{\pm.004}$ | $0.628^{\pm.004}$ | $0.743^{\pm.004}$ | $3.015^{\pm.017}$ | $11.417^{\pm.097}$ |
| Fg-T2M [38] | $0.571^{\pm.047}$ | $0.418^{\pm.005}$ | $0.626^{\pm.004}$ | $0.745^{\pm.004}$ | $3.114^{\pm.015}$ | $10.93^{\pm.083}$ |
| AttT2M [16] | $0.870^{\pm.039}$ | $0.413^{\pm.006}$ | $0.632^{\pm.006}$ | $0.751^{\pm.006}$ | $3.039^{\pm.021}$ | $10.96^{\pm.123}$ |
| DiverseMotion [43] | $0.468^{\pm.098}$ | $0.416^{\pm.005}$ | $0.637^{\pm.008}$ | $0.760^{\pm.011}$ | $2.892^{\pm.041}$ | $10.873^{\pm.101}$ |
| ParCo [42] | $0.453^{\pm.027}$ | $0.430^{\pm.004}$ | $0.649^{\pm.007}$ | $0.772^{\pm.006}$ | $2.820^{\pm.028}$ | $10.95^{\pm.094}$ |
| MMM [21] | $0.429^{\pm.019}$ | $0.381^{\pm.005}$ | $0.590^{\pm.006}$ | $0.718^{\pm.005}$ | $3.146^{\pm.019}$ | $10.633^{\pm.097}$ |
| MoMask [17] | $0.204^{\pm.011}$ | $0.433^{\pm.007}$ | $0.656^{\pm.005}$ | $0.781^{\pm.005}$ | $2.779^{\pm.022}$ | - |
| Ours | $\mathbf{0.143}^{\pm.004}$ | $\mathbf{0.445}^{\pm.006}$ | $\mathbf{0.671}^{\pm.006}$ | $\mathbf{0.797}^{\pm.005}$ | $\mathbf{2.711}^{\pm.024}$ | $10.918^{\pm.090}$ |

Table 1: Evaluation on the HumanML3D dataset (upper half) and the KIT-ML dataset (lower half).

are constructed from 2 convolutional residual blocks with a downscale of 4. The transformers in our model are all set to have 6 layers, 6 heads, and 384 latent dimensions. The parameter $\alpha$ is set to 1 and $N$ is set to 10. More details are described in the supplementary material.

## 4.3 Evaluation

We evaluate both motion quantization and motion generation in our framework. The evaluation protocol strictly adheres to the standards established by previous methods [5, 14, 17, 21], using the same evaluator and metric calculation. Each experiment is repeated 20 times and the results are reported alongside a 95% statistical confidence interval.

| Methods | HumanML3D | | KIT-ML | |
|---|---|---|---|---|
| | FID ↓ | MPJPE ↓ | FID ↓ | MPJPE ↓ |
| TM2T [13] | 0.307 | 230.1 | - | - |
| M2DM [15] | 0.063 | - | 0.413 | - |
| T2M-GPT [14] | 0.070 | 58.0 | 0.472 | - |
| MoMask [17] | $0.019^{\pm.000}$ | $29.5^{\pm.0}$ | $0.112^{\pm.002}$ | $37.2^{\pm.1}$ |
| Ours | $\mathbf{0.005}^{\pm.000}$ | $\mathbf{13.8}^{\pm.0}$ | $\mathbf{0.019}^{\pm.001}$ | $\mathbf{17.4}^{\pm.1}$ |

Table 2: Evaluation of motion quantization on HumanML3D dataset and KIT-ML dataset. MPJPE is measured in millimeters.

**Evaluation of Motion Quantization.** The core idea of our framework starts from the spatial-temporal 2D joint quantization. Therefore, we first evaluate our VQVAE against that of previous quantization-based methods. For a fair comparison, we temporarily configure our codebook to the

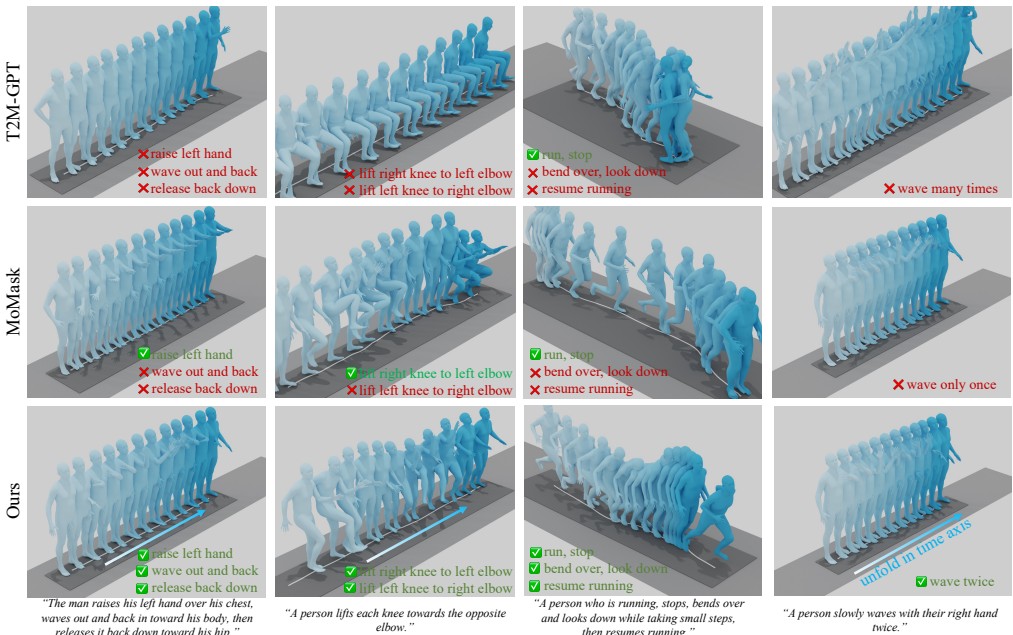

Figure 4: Qualitative results on the test set of HumanML3D. The color from light blue to dark blue indicates the motion sequence order. An arrow indicates this sequence is unfolded in the time axis.

same size as the previous methods [17], i.e., $512 \times 512$. As reported in Table 2, the FID and MPJPE accuracy of our method exceeds previous methods by a large margin, in both the HumanML3D and the KIT-ML datasets. This reveals that the difficulty of quantizing an individual joint is much smaller than quantizing a whole-body pose.

**Evaluation of Motion Generation.** We then compare our model to previous text-to-motion works, including both the continuous regression-based methods and the discrete quantization-based methods. From the results reported in Table 1, our method outperforms all previous methods on both the HumanML3D and the KIT-ML datasets, which demonstrates the effectiveness of our method. Specifically, the FID is decreased by $26.6\%$ on HumanML3D and is decreased by $29.9\%$ on KIT-ML. In addition, the R-precision even significantly surpasses the ground truth. From the qualitative results displayed in Figure 4, we also see that our method generates motions that are better matched with the input texts compared to previous methods.

### 4.4 Ablation Study

**Joint 2D VQ.** To verify the efficacy of the proposed framework, we first establish a baseline model incorporating a 1D motion VQVAE and a 1D token prediction transformer. Then we change the VQVAE to our 2D joint VQ-VAE, examining its impact on both motion auto-encoding and motion generation capabilities. According to the outcomes presented in Table 2, we observed a substantial enhancement in auto-encoding performance. However, as indicated

|  | FID ↓ | Top1 ↑ |
|---|---|---|
| Baseline | 0.108 | 0.501 |
| + 2D VQ | 0.194 | 0.493 |
| + 2D Masking | 0.054 | 0.516 |
| + 2D Position Encoding | 0.047 | 0.521 |
| + S&T Attn | 0.033 | 0.529 |

Table 3: Ablation study on HumanML3D dataset.

in Table 3, the motion generation performance deteriorates. This decline can be attributed to the considerable increase in the number of tokens, which renders the 1D masking strategy and 1D transformer no longer adequate.

**2D Masking.** We then change the 1D masking to temporal-spatial 2D token masking, which leads to a significant performance improvement, as depicted in Table 3. This demonstrates the necessity of our 2D masking strategy, as discussed in Section 3.4.

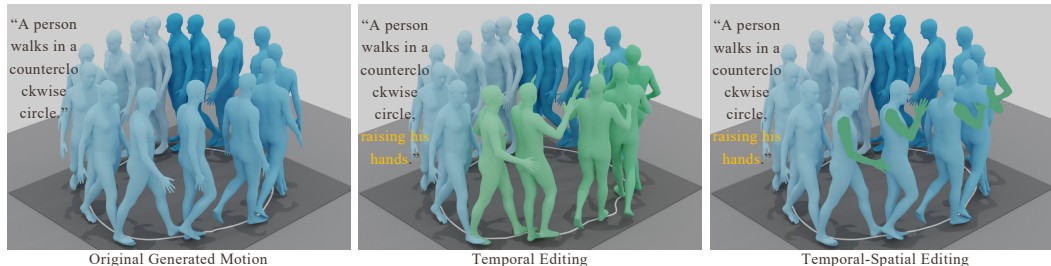

Figure 5: Motion Editing. The edited regions are indicated in green.

**2D Position Encoding.** With the 2D VQ, the attention is calculated between 2D tokens, but it is not a full version of the 2D attention. We upgrade this to spatial-temporal 2D attention by integrating the 2D position encoding, which provides both the spatial location and temporal location information. The performance improvement brought by this encoding is recorded in Table 3.

**Joint Spatial and Temporal Attention.** As discussed in Section 3.5.1, the mix of spatial and temporal information is complex for the bidimensional attention performed on a long sequence. Here we inspect the benefits brought by the individual joint spatial and temporal attention mechanism. From the results in Table 3, this design further improves the performance of our framework.

### 4.5 Motion Editing / Inpainting

Due to generative mask modeling, our method is capable of not only generating motions from texts, but also editing a motion by masking the tokens of any locations in both the temporal and spatial dimensions. As shown in Figure 5, we first mask out the motion in the temporal dimension and then generate the motions corresponding to the masked locations according to a different text prompt, resulting in a motion in which only the masked frames are edited. Subsequently, we perform similar editing in the spatial dimension, which results in a motion where only the masked joints of masked frames are edited. This editing mechanism could also be applied to inpainting a motion sequence or connecting two motion sequences by generating missing motions.

## 5 Conclusion

This paper proposes to quantize each individual joint to one vector, generating a spatial-temporal 2D token mask for motion quantization, which reduces the approximation error in quantization, reserves the spatial information between different joints for subsequent processing, and enables the 2D operators widely used in 2D images. Then the temporal-spatial 2D masking and spatial-temporal 2D attention are proposed to leverage the spatial-temporal information between joints for motion generation. Extensive experiments demonstrate the efficacy of the proposed method.

### 5.1 Limitations and future work.

**Quantization.** Despite the improvement of our method in the quantization, there is still the approximation error, which limits the motion generation. This should be further improved in the future network design, e.g., employing the coarse-to-fine technology. Another important point is the restricted size of the current human motion dataset. In the image pretraining, a much larger dataset is utilized to achieve decent quantization results. Similarly, a large dataset for pretraining an accurate human motion quantizer is also needed in future work.

**Generation.** We follow the mask-and-generation for motion generation in this work. The masking strategy used in the current framework is borrowed from Bert [24]. On the contrary, some other methods follow the auto-regressive generation and also achieve good performance. In the future, one promising direction is combining these two technologies.

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

# A  Appendix

## A.1  Evaluation Metrics

We strictly follow previous methods [5, 14, 17] to evaluate our model. We use the evaluator proposed in [5] to calculate the text-motion align feature of the generated motion, ground-truth motion, and text as $\mathbf{f}_{\text{gen}}, \mathbf{f}_{\text{gt}}, \mathbf{f}_{\text{text}}$ Then we evaluate the features with the following metrics:

**Frechet Inception Distance (FID)**  is an objective metric calculating the distance between features extracted from the generated motion and the ground-truth motion, as

$$\text{FID} = ||\mu_{\text{gen}} - \mu_{\text{gt}}||^2 - Tr(Cov_{\text{gen}} + Cov_{\text{gt}} - 2(Cov_{\text{gen}}Cov_{\text{gt}})^{\frac{1}{2}}), \tag{11}$$

where $\mu_{\text{gen}}$ and $\mu_{\text{gt}}$ are the mean of $\mathbf{f}_{\text{gen}}$ and $\mathbf{f}_{\text{gt}}$, $Cov$ denotes the covariance matrix, and $Tr$ denotes the trace of a matrix. The features are extracted using the models following previous methods [5].

**R-Precision (Top-1, Top-2, Top-3)**  measures the matching between the motion and the text. For each generated motion, its ground-truth text description and 31 randomly selected mismatched descriptions from the test set form a description pool. Then this metric is calculated by computing and ranking the Euclidean distances between the motion feature and the text feature of each description in the pool. Then the average accuracy is counted at Top-1, Top-2, and Top-3 places. The ground truth entry falling into the top-k candidates is treated as successful retrieval.

**MultiModal-Dist (MM-Dist)**  measures the distance between the text feature and the motion feature. This metric is calculated by the average Euclidean distance between these two features, as

$$\text{MM-Dist} = ||\mathbf{f}_{\text{gen}} - \mathbf{f}_{\text{text}}||^2 \tag{12}$$

**Diversity**  measures the variance of the whole motion sequences across the dataset. It randomly samples $N_{\text{diver}}$ pairs of motion, where each pair is denoted by $\mathbf{f}^{i,1}$ and $\mathbf{f}^{i,2}$. Then the diversity is computed as

$$\text{Diversity} = \frac{1}{N_{\text{diver}}} \Sigma_{i=1}^{N_{\text{diver}}} ||\mathbf{f}^{i,1} - \mathbf{f}^{i,2}||, \tag{13}$$

where $N_{\text{diver}}$ is set to 300.

## A.2  Computational Overhead

We test the computational overhead of different methods on an NVIDIA 4090 GPU and report the average inference time per sentence in Table 4. Although our method does not achieve the shortest inference time, the increase in computational overhead is not significant. Overall, the computational overhead of our method is comparable to that of mainstream methods.

| Method | Average Inference Time per Sentence |
|---|---|
| MLD | 134 ms |
| MotionDiffuse | 6327 ms |
| MDM | 10416 ms |
| T2M-GPT | 239 ms |
| MoMask | 73 ms |
| Ours | 181 ms |

Table 4: Computational overhead of different methods.

## A.3  More Results

### A.3.1  Motion Quantization Evaluation on More Datasets

To further assess the spatial-temporal 2D motion quantization of our method, we perform the evaluation on a recent larger dataset, Motion-X [57]. This dataset collects 81.1K motion sequences from various public datasets and some self-captured videos. We convert their data to the 263-format

| Methods | MPJPE $\downarrow$ | FID $\downarrow$ | Top1 $\uparrow$ | Top2 $\uparrow$ | Top3 $\uparrow$ | MM-Dist $\downarrow$ | Diversity $\rightarrow$ |
|---|---|---|---|---|---|---|---|
| Ground Truth | - | - | $0.511^{\pm.003}$ | $0.703^{\pm.003}$ | $0.797^{\pm.002}$ | $2.974^{\pm.008}$ | $9.503^{\pm.065}$ |
| MoMask [17] | $29.5^{\pm.0}$ | $0.019^{\pm.000}$ | $0.508^{\pm.003}$ | $0.701^{\pm.002}$ | $0.795^{\pm.002}$ | $2.999^{\pm.006}$ | $9.565^{\pm.080}$ |
| Ours | $13.8^{\pm.0}$ | $0.005^{\pm.000}$ | $0.512^{\pm.002}$ | $0.704^{\pm.002}$ | $0.798^{\pm.002}$ | $2.978^{\pm.006}$ | $9.501^{\pm.076}$ |

(a)

| Methods | MPJPE $\downarrow$ | FID $\downarrow$ | Top1 $\uparrow$ | Top2 $\uparrow$ | Top3 $\uparrow$ | MM-Dist $\downarrow$ | Diversity $\rightarrow$ |
|---|---|---|---|---|---|---|---|
| Ground Truth | - | - | $0.424^{\pm.005}$ | $0.649^{\pm.006}$ | $0.779^{\pm.006}$ | $2.788^{\pm.012}$ | $11.080^{\pm.097}$ |
| MoMask [17] | $37.2^{\pm.2}$ | $0.112^{\pm.001}$ | $0.417^{\pm.006}$ | $0.645^{\pm.007}$ | $0.769^{\pm.004}$ | $2.786^{\pm.016}$ | $10.811^{\pm.130}$ |
| Ours | $17.4^{\pm.1}$ | $0.019^{\pm.001}$ | $0.423^{\pm.004}$ | $0.649^{\pm.005}$ | $0.779^{\pm.005}$ | $2.757^{\pm.012}$ | $11.056^{\pm.069}$ |

(b)

Table 5: Evaluation of motion quantization on (a) Humanml3D dataset and (b) KIT-ML dataset. MPJPE is measured in millimeters.

| Methods | MPJPE $\downarrow$ | FID $\downarrow$ | Top1 $\uparrow$ | Top2 $\uparrow$ | Top3 $\uparrow$ | MM-Dist $\downarrow$ | Diversity $\rightarrow$ |
|---|---|---|---|---|---|---|---|
| Ground Truth | - | - | $0.420^{\pm.002}$ | $0.631^{\pm.001}$ | $0.754^{\pm.002}$ | $2.800^{\pm.003}$ | $10.100^{\pm.083}$ |
| MoMask [17] | $111^{\pm.0}$ | $0.081^{\pm.001}$ | $0.396^{\pm.002}$ | $0.604^{\pm.002}$ | $0.725^{\pm.002}$ | $2.955^{\pm.003}$ | $9.837^{\pm.103}$ |
| Ours | $48.7^{\pm.0}$ | $0.011^{\pm.000}$ | $0.417^{\pm.002}$ | $0.627^{\pm.002}$ | $0.750^{\pm.001}$ | $2.832^{\pm.003}$ | $10.113^{\pm.094}$ |

Table 6: Evaluation of motion quantization on Motion-X dataset. MPJPE is measured in millimeters.

aligned with HumanML3D [5] following the official code of Motion-X. Since there is no official text-motion-matching model, we only evaluate the motion quantization part of our framework. We follow [5] to train the text & motion feature extractors for evaluation. The results are reported in Table 6. Besides, the complete results of quantization evaluation on HumanML3D and KIT-ML are also reported in Table 5. The results indicate that our method is capable of quantizing continuous motions with significantly reduced loss. This enhancement allows our transformer to work with motion tokens with smaller approximation errors, thus raising the upper limit of motion generation quality.

### A.3.2 More Qualitative Results

We display more qualitative results in Figure 6 and Figure 7. From the generated motions, we see that our method could generate complex and interesting motion sequences according to the text prompts.

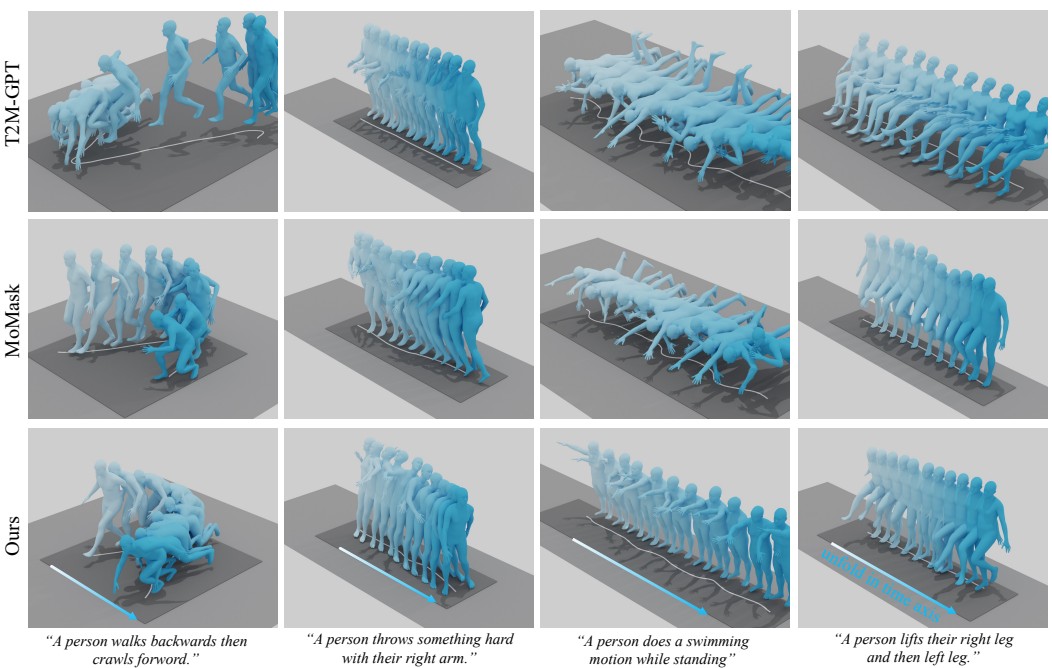

Figure 6: More qualitative results of our method are presented. The color from light blue to dark blue indicates the motion sequence order. An arrow indicates this sequence is unfolded in the time axis.

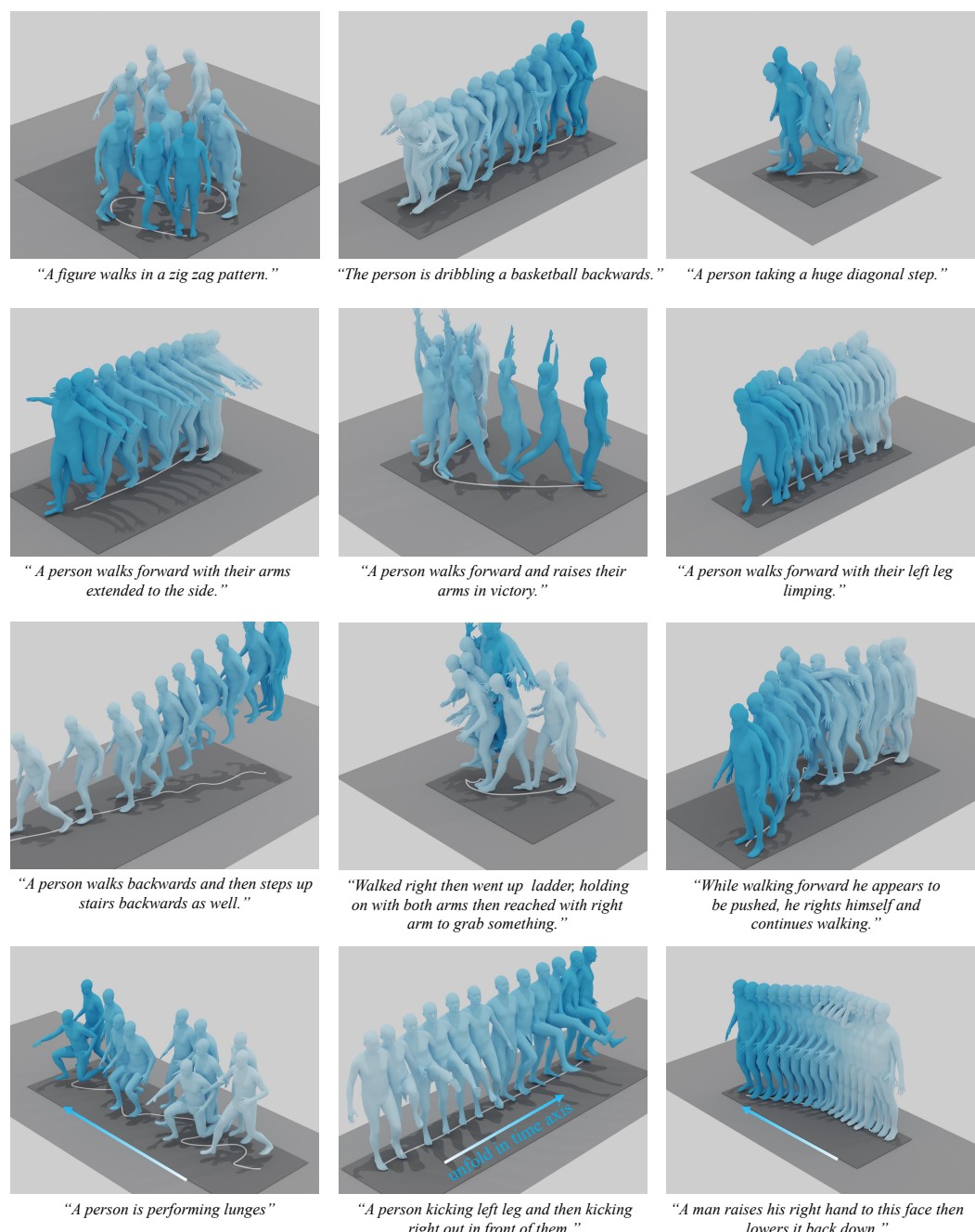

*"A figure walks in a zig zag pattern."*  *"The person is dribbling a basketball backwards."*  *"A person taking a huge diagonal step."*

*" A person walks forward with their arms extended to the side."*  *"A person walks forward and raises their arms in victory."*  *"A person walks forward with their left leg limping."*

*"A person walks backwards and then steps up stairs backwards as well."*  *"Walked right then went up ladder, holding on with both arms then reached with right arm to grab something."*  *"While walking forward he appears to be pushed, he rights himself and continues walking."*

*"A person is performing lunges"*  *"A person kicking left leg and then kicking right out in front of them."*  *"A man raises his right hand to this face then lowers it back down."*

Figure 7: More qualitative results of our method are presented. The color from light blue to dark blue indicates the motion sequence order. An arrow indicates this sequence is unfolded in the time axis.

