# OpenReview forum: "MoGenTS: Motion Generation based on Spatial-Temporal Joint Modeling"
_NeurIPS.cc/2024/Conference — NeurIPS 2024 poster_

### Official Review · Reviewer_1rhT · 2024-06-25

**Soundness:** 3
**Presentation:** 3
**Contribution:** 3
**Rating:** 6
**Confidence:** 4

**Summary:**

This paper present a new method for text to motion generation. In this method the human motion is represented as 2D tokens in a codebook. This allow the authors to apply 2D operation on 3D motions and use a 2D masking strategy. The architecture is composed of a VAE to learn the codebook and of a Transformer to learn the relation between CLIP embedding of the text input and the corresponding codebook tokens using spatial-temporal attention. With this method the authors outperforms state of the art approaches quantitatively and qualitatively. An ablation study shows the effect of each component.

**Strengths:**

The paper is clear and detailed.

The mixed use of codebook, masking ans spatial-temporal transformer is interesting.

The method outperforms the state of the art quantitatively and qualitatively.

The ablation shows well the effect of each component.

**Weaknesses:**

The Figure 3 b is not very clear It seems that the CLIP embedding is concatenated to the flattened motion embedding and then positional encoding is applied while the text say that the CLIP embedding is added after positional encoding. I also don't understand why positional encoding is added twice, one on the flattened vector and one time on the matrix.

An ablation to show that this concatenation of token and text is the better than for example cross attention would have been welcome. Another interesting ablation would have been to see the performance of the model when computing the temporal and spatial attention in parallel instead of sequentially.

It is not clear whether P is added only during attention or also on the inputs like base transformer. There is also no explanation as to why add positional encoding after computing the attention matrix.

On several metric the ground truth is beaten but the paper does not provide explanation fro this.

The paper does not describe how FID features are extracted ?

The classifier free guidance should be mentioned in the main paper not just in the appendix. It is an important component.

Regarding the motion editing figure : why is only one hand being raised with the Temporal-Spatial Editing while the temporal editing results shows both hands being raised. Since the plural is used in the prompt this would indicate that temporal editing is better.

**Questions:**

It should be mentioned somewhere that j^i_t contains the 3 dimension of joint i.
A user study would have been nice. Metrics are difficult to use on these more complex actions.
It might be better to mention clip in the overview instead of waiting fro section 3.5.

**Limitations:**

The limitations are very briefly addressed.

---

> ### Author Rebuttal · Authors · 2024-08-04
>
> We thank the reviewer for the valuable feedback and constructive suggestions! We hope our responses adequately address the following questions raised about our work. Please let us know if there is anything we can clarify further.
>
> **1. Clarification of positional encoding.**
>
> Sorry for the confusion caused. We will clarify this in the revision.
>
> - The positional encoding is performed after adding the CLIP embedding. We will revise the manuscript in the revision.
> The reason why positional encoding is added twice is that we want to reinforce the network's awareness of the spatial-temporal 2D structure. Adding positional encoding once should also work.
>
> - The positional encoding P is added on the inputs like the base transformer.
>
> - Adding positional encoding after computing the attention matrix is because we want to reinforce the network's awareness of the spatial-temporal 2D structure before the next attention computing.
>
>
> **2. Ablation of cross attention of token and text and computing the temporal and spatial attention in parallel.**
>
> Thanks for this constructive advice. We perform the ablation study as suggested and present the results in Table 2.
> From the results, we can see that the cross attention of token and text or computing the temporal and spatial attention in parallel performs similarly to our original framework.
>
> | **Setup**          |     **FID**   |      **Top1**   |
> |-------------------|   :------------------:| :------------------:|
> | Cross attention of token and text     |  0.034  | 0.521  |
> | Computing the temporal and spatial attention in parallel     |  0.035  | 0.522  |
> | Ours     |  0.033  | 0.529  |
>
>
> **3. Why ground truth is beaten.**
>
> The ground truth is beaten on the metrics about the text-motion matching, so we think this is because the text label is not perfectly annotated.
> We will add this explanation in the revision.
>
>
> **4. FID features extract.**
>
> Following previous methods, we use the same models proposed in T2M [5] to extract the FID features. We will add this description in the revision.
>
> **5. Classifier free guidance.**
>
> We will mention this in the main paper in the revision.
>
>
> **6. Motion editing figure.**
>
> Sorry for the misleading.
> In Figure 5, the Temporal-Spatial Editing is just to show that we could edit only specific joints while keeping other joints fixed. This level of granular control is hard to realize in previous methods that encode all joints to one token.
>
> **7. User study.**
>
> Thanks for this constructive advice. For the user study, we select 10 texts and display the generated results of T2M-GPT, MoMask, and Ours. Due to the limited time, we only collect the study from 14 users, resulting in 140 total votes. Of these, 102 votes favor our results, 33 votes prefer MoMask, and 5 votes prefer T2M-GPT, as is shown in Figure 1 in the uploaded PDF file.
>
>
> **8. Other questions.**
>
> - We will mention that $j^i_t$ denotes a 3D rotation of joint i on time t in the revision.
> - We will mention CLIP in the method overview in the revision.
> - Limitations: The brevity is due to the limited space. We will try to include a more extensive discussion in the revision.

---

> ### Comment · Reviewer_1rhT · 2024-08-09
> **Rating after rebuttal**
>
> The authors clarified the few thing that weren't clear to me. The user study is appreciated. I keep my original weak accept rating.
> It seems that the values reported inside the user study graph are wrong (102,73%).

---

> > ### Author Response · Authors · 2024-08-13
> > **Thanks for the reviewer's feedback**
> >
> > Thanks for the reviewer's feedback. We are pleased to see that our response has provided clarification on some points.
> >
> > Sorry for the confusion. When we state (102, 73%), we mean that there are 102 votes, which represent 73% of the total votes. We will revise this for clarity.

---

### Official Review · Reviewer_SBXK · 2024-07-02

**Soundness:** 4
**Presentation:** 2
**Contribution:** 3
**Rating:** 6
**Confidence:** 4

**Summary:**

This paper proposes an approach for text-conditioned motion generation. A common practice in this area is to use a quantized representation of human motion obtained with a VQ-VAE. However, most prior works represent the full body by a single token, which makes accurate reconstruction complicated.

In this work, the authors propose a new way of quantizing human motion: a single token is associated with a single joint. Then, the motion can be represented as a 2D grid of indices corresponding to spatial and temporal dimensions. Using this new representation, this paper proposes to generate motions using masked generative modeling.

In summary, the contributions of this paper are:
- A new quantization of the human motion, representing each joint in a 2D map of tokens.
- A masking strategy allowing the leverage of spatiotemporal information preserved by the proposed quantization.
- A masked generative modeling strategy to generate new motions conditioned on text input.

**Strengths:**

I would say that the main strength of this paper is not the novelty: masked generative modeling was already used for human motion generation [17]. However, this paper brings new components that make a lot of sense and seem to greatly impact the results. The bottleneck of prior works (1 pose = 1 token) is well-identified, and the proposed quantization strategy addresses this problem effectively by associating a token to each joint. In addition to improving the reconstruction after quantization, this representation proves useful as it preserves the spatiotemporal structure of the motion. Carefully designed operations (2D token masking, spatial-temporal motion Transformer) benefit from the proposed representation despite its higher dimension.

Another strength of this paper is the evaluation. The comparisons follow the standard procedures and seem totally fair to other methods. Providing confidence intervals by running experiments multiple times is an excellent practice. Even for the evaluation of the quantization in Table 2, I find it very good that the authors decreased the codebook size of the introduced model for fair comparison with other methods. In addition to the 2 datasets widely used for comparisons, the appendix provides results on numerous datasets, which is appreciated to evaluate the model's generalization capability. The ablations are also satisfying, as they allow to evaluate the impact of the main introduced components.

**Weaknesses:**

The main weakness of this paper is that it is difficult to understand the quantization of human motion:
- L76 "each joint is quantized to an individual code of a VQ book": From my understanding, with the residual quantization, each joint is quantized to a sequence of indices; the final code is the sum of codes corresponding to those indices and associated codebooks.
- Equation 1: Given L159, it seems that one joint in the input is converted to one token. Equation 1 suggests the same (the input of the encoder would be of dimension 3). And then L272, "Both the encoder and decoder are constructed from 2 convolutional residual blocks with a downscale of 4," so I really do not understand at all. Is there a spatiotemporal reduction?
- Equation 2: This does not correspond to residual quantization. Maybe it is meant to simplify the understanding, but I find it very confusing.

Globally, it is very difficult to understand how the method works until we reach section 3.5.2. For instance, until then, I did not understand the notion of a 2D map since the residual quantization would have made the grid 3-dimensional. I also wondered how the masking could encompass the depth of the quantization.

Other minor issues include:
- L125: It would be better to mention methods that represent a single pose (or human) with multiple tokens [a,b].
- The presentation of Table 1 is not optimal. Giving the dataset in the table instead of the caption would be more clear (like in [17]). Also, why are there no bold results for diversity? It may look like this is because some other methods have better results.



[a] Geng, Z., Wang, C., Wei, Y., Liu, Z., Li, H., & Hu, H. (2023). Human pose as compositional tokens. In Proceedings of the IEEE/CVF Conference on Computer Vision and Pattern Recognition (pp. 660-671).

[b] Fiche, G., Leglaive, S., Alameda-Pineda, X., Agudo, A., & Moreno-Noguer, F. (2023). VQ-HPS: Human Pose and Shape Estimation in a Vector-Quantized Latent Space. arXiv preprint arXiv:2312.08291.

**Questions:**

- How does the quantization work? I would like to understand if there is a spatial-temporal reduction, as the information in the paper seems contradictory.
- From my understanding in Section 3.3, there is no information about which joint is processed in the encoder (so the encoder processes in the same manner every joint). Can the VQ-VAE be considered a quantization of the 3D space as a learned grid?
- From Figure 2 and other explanations, the pose seems flattened to correspond to a column in the 2D map. How are the joints organized so that the spatial structure of the pose is preserved? Is there a topological ordering?

**Limitations:**

The section about the limitations is very short. I think that it could be improved by proposing more solutions to the quantization problem and other research perspectives. Otherwise, it sounds like the problem of motion generation is now completely solved.

The authors say that this work has no societal impact at all. I would agree that the impact on society is very limited.

---

> ### Author Rebuttal · Authors · 2024-08-04
>
> We thank the reviewer for the valuable feedback and constructive suggestions! We hope our responses adequately address the following questions raised about our work. Please let us know if there is anything we can clarify further.
>
> **1. Clarification of the quantization.**
>
> Sorry for the confusion caused. We will clarify this in the revision. In this paper, we do not intend to detail the residual quantization, since we simply follow established methods to employ this technique, as noted in L169. Thus for simplicity, we describe our method using only single-level quantization without incorporating a residual structure.
> Therefore,
> - L76: Here we only describe the single-level quantization without the residual structure.
> - Spatiotemporal reduction: In equation 1 within the method section, we describe our methodology that does not include spatiotemporal reduction.
> However, our experimental results indicate that appropriate spatiotemporal reduction can decrease computational load without significantly impacting performance.
> So we perform a spatio-temporal reduction in the experiments and describe this in L272 in the implementation details.
> - Equation 2: Yes, here we only describe the single-level quantization without the residual structure.
> - Masking: The masking is only performed on the base-level. The prediction of the tokens in residual levels is based on the previous level without masking, following the previous method MoMask.
>
> **2. Encoder processes in the same manner every joint.**
>
> Yes, the encoder processes every joint in the same manner. The 2D joint map, with dimensions T×J×D (where T represents the sequence length, J denotes the number of joints, and D indicates the feature dimension), is processed by the 2D convolutional networks, similar to image processing techniques.
> After the encoder, the quantization (either single-level or residual) begins to quantize the encoded vectors.
>
>
> **3. Spatial structure.**
>
> The order of the joints is fixed in both training and inference, so we hope the network can learn the spatial structure of the flattened joints in the training, with the 2D positional encoding.
>
> **4. Minor issues and limitations.**
>
> We will mention [a, b] in the revision.
>
> Table 1: Giving the dataset in the caption is due to space constraints. We will attempt to revise this. No bold results for diversity is because it is hard to say which one is better for this metric.
>
> Limitations: The brevity is due to the limited space. We will try to include a more extensive discussion on quantization in the revision.

---

> > ### Comment · Reviewer_SBXK · 2024-08-09
> >
> > Thanks to the authors for an insightful rebuttal addressing most of my concerns. I still lean towards accepting this paper.
> >
> > I have a doubt about the **spatial structure**. I understand that the joints' ordering is the same at training and inference and that there is a positional encoding. My question was more about the order of the joints once flattened: does it follow an order that preserves the structure of the skeleton (for instance, left shoulder -> left elbow -> left wrist, ...), or is the spatial information exclusively in the positional encoding?
> >
> > For the lack of space in the caption of Table 1 and the limitations, I would suggest moving the limitations to the annexes.

---

> ### Author Response · Authors · 2024-08-12
>
> Thanks for the reviewer's feedback. We are pleased to see that our response has addressed most of the concerns.
>
> Regarding the spatial structure, the flattened order follows the HumanML3D dataset and is as follows: 'pelvis', 'right_hip', 'left_hip', 'spine1', 'right_knee', 'left_knee',  'spine2', 'right_ankle', 'left_ankle', 'spine3', 'right_foot', 'left_foot', 'neck', 'right_collar', 'left_collar', 'head', 'right_shoulder', 'left_shoulder', 'right_elbow', 'left_elbow', 'right_wrist', 'left_wrist'. These joints are arranged in order of proximity to the pelvis joint, from nearest to furthest.
>
> Sure, we will move the limitation to the annexes and discuss more.

---

### Official Review · Reviewer_g7ot · 2024-07-13

**Soundness:** 3
**Presentation:** 3
**Contribution:** 3
**Rating:** 5
**Confidence:** 5

**Summary:**

This paper proposes a novel approach to human motion generation by quantizing each joint into individual vectors, rather than encoding the entire body pose into one code. The key contributions are: (1) It quantizes each joint separately to preserve spatial relationships and simplify the encoding process. Then the motion sequence are organized into a 2D token map, akin to 2D images, allowing the use of 2D operations like convolution, positional encoding, and attention mechanisms. (2) It introduces a spatial-temporal 2D joint VQVAE to encode motion sequences into discrete codes and employs a temporal-spatial 2D masking strategy and a spatial-temporal 2D transformer to predict masked tokens.

**Strengths:**

1. The paper introduces a novel joint-level quantization approach, addressing the complexity and spatial information loss issues seen in whole-body pose quantization.By organizing motion sequences into a 2D token map, the method takes advantage of powerful 2D image processing techniques, enhancing feature extraction and motion generation.
2. The integration of 2D joint VQVAE, temporal-spatial 2D masking, and spatial-temporal 2D attention forms a robust framework that effectively captures spatial-temporal dynamics in human motion.
3. Extensive experiments demonstrate the method's efficacy, outperforming previous state-of-the-art methods on key datasets.
4. The paper is well-written and easy to understand. The supp. mat. video provides comparisons with MoMask.

**Weaknesses:**

1. Although the overall idea of joint-level quantization is interesting, I still have the concern of computational overhead. While motion representation is typically lightweight, the use of 2D code maps and spatial-temporal attention can introduce significant computational overhead, similar to image data processing. It would be beneficial to compare the inference speed of mainstream methods (e.g., MoMask, T2M-GPT, MLD, etc) to show that the state-of-the-art performance is achieved with comparable computational costs.
2. The experiments are conducted on relatively small datasets (HumanML3D and KIT-ML). To better validate the effectiveness of the proposed method, experiments on larger-scale datasets, such as Motion-X, would be advantageous.

**Questions:**

Please see the weaknesses. Overall, the idea is interesting. My major concern is the extra computational cost of this method, which could be much larger than previous methods with body-level VQ-VAE yet it is not investigated in the paper.

**Limitations:**

This paper has discusses the limitation of this paper: the approximation error in VQ-VAE and a larger dataset for training more accurate VQ-VAE.
I think there is no potential negative societal impact in this paper.

---

> ### Author Rebuttal · Authors · 2024-08-03
>
> We thank the reviewer for the valuable feedback and constructive suggestions! We hope our responses adequately address the following questions raised about our work. Please let us know if there is anything we can clarify further.
>
> **1. Computational overhead of the proposed method.**
>
> Thanks for this constructive advice. We test the computational overhead of different methods on an NVIDIA 4090 GPU and report the average inference time per sentence in the Table 1.
> Although our method does not achieve the shortest inference time, the increase in computational overhead is not significant.
> Overall, the computational overhead of our method is comparable to that of mainstream methods.
>
> | **Methods**          |     **Average Inference Time per Sentence**   |
> |-------------------|   :------------------:|
> | MLD     |  134 ms  |
> | MotionDiffuse     |  6327 ms  |
> | MDM     |  10416 ms  |
> | T2M-GPT     |  239 ms  |
> | MoMask     |  73 ms  |
> | Ours     |  181 ms  |
>
>
>
> **2. Experiments on Motion-X.**
>
> Sorry for the lack of clarity. Actually, we have conducted the motion quantization experiments on the Motion-X dataset, as in Appendix A.3.1 or the below table. From the quantization results, our method also works well in the larger dataset, Motion-X.
> Due to the fact that there is no official text-motion-feature model, it is hard to evaluate the text-to-motion generation performance of our method. We have trained our own text-motion-feature model following HumanML3D, but it does not work well. We will clarify this in the revision, and also continue our attempts to evaluate our method on the Motion-X dataset.
>
> | **Methods** | **MPJPE** | **FID** | **Top1** | **Top2** | **Top3** | **MM-Dist** | **Diversity** |
> |-------------------|   :------------------:| :------------------:|  :------------------:| :------------------:|  :------------------:| :------------------:| :------------------:|
> | Ground Truth     |  -  | -  | 0.420 | 0.631 | 0.754 | 2.800 | 10.100
> | Momask     |  111  | 0.081  | 0.396 | 0.604 | 0.725 | 2.955 | 9.837
> | Ours     |  48.7  | 0.011  | 0.417 | 0.627 | 0.750 | 2.832 | 10.113

---

> > ### Comment · Reviewer_g7ot · 2024-08-12
> >
> > After I carefully read other reviews and the author rebuttal, I think this paper proposes an effective and efficient method for motion generation. My initial concerns about the efficiency and the results on Motion-X has also been resolved in the author rebuttal. I will keep my original rating and leaning to accept this paper.

---

> > > ### Author Response · Authors · 2024-08-14
> > > **Thanks for the reviewer's feedback**
> > >
> > > Thanks for the reviewer's feedback. We are pleased to see that our response has addressed the concerns. If there are no further concerns, please also consider raising the rating. Many thanks!

---

### Author Rebuttal · Authors · 2024-08-05

We would like to express our sincere gratitude to all the reviewers for their time and their valuable feedback. We deeply appreciate their recognition of our work, such as

**Reviewer g7ot:**
"a novel approach",
"a robust framework that effectively captures spatial-temporal dynamics",
"Extensive experiments demonstrate the method's efficacy, outperforming previous state-of-the-art methods".

**Reviewer SBXK:**
"this paper brings new components that make a lot of sense and seem to greatly impact the results"

**Reviewer 1rhT:**
"The mixed use of codebook, masking and spatial-temporal transformer is interesting",
"The method outperforms the state of the art quantitatively and qualitatively"

In the following, we address each reviewer’s comments point by point, and attach some figures in the uploaded PDF file.
We hope our responses adequately address the questions raised about our work. Please let us know if there is anything else we can clarify further.

---

### Decision · Program_Chairs · 2024-09-25

**Decision:**

Accept (poster)

**Comment:**

This paper presents work on 3d pose human motion generation.  The main contribution is quantization using body joints rather than entire poses.  This contribution is incorporated into a framework that includes masking and attention, leading to good quality results with respect to the state of the art.

The reviewers appreciated the exploration of joint-level encoding as a means to more effective quantization-based motion generation.  The overall framework is sensible, and the experimentation thorough.  Concerns were raised over computational costs as well as some details on clarity.  In the subsequent author response / discussion the computational costs and clarity issues were largely resolved.

Overall, the paper's core contributions are very focused on encoding for human motion generation.  The results are convincing and the paper is solidly executed.  I am not sure whether the claims regarding spatial relationships being lost in entire body pose encoding is accurate -- the whole pose is encoded, so spatial relationships are present; this seems more about compositionality.  However, this is a relatively minor point and overall the paper is a solid contribution and advances 3d human motion generation.  As such, it is recommended for acceptance in NeurIPS.